# Conformal Normalization in Recurrent Neural Network of Grid Cells

## Abstract

Grid cells in the entorhinal cortex of the mammalian brain exhibit striking hexagon firing patterns in their response maps as the animal (e.g., a rat) navigates in a 2D open environment. The responses of the population of grid cells collectively form a vector in a high-dimensional neural activity space, and this vector represents the self-position of the agent in the 2D physical space. As the agent moves, the vector is transformed by a recurrent neural network that takes the velocity of the agent as input. In this paper, we propose a simple and general conformal normalization of the input velocity for the recurrent neural network, so that the local displacement of the position vector in the high-dimensional neural space is proportional to the local displacement of the agent in the 2D physical space, regardless of the direction of the input velocity. Our numerical experiments on the minimally simple linear and non-linear recurrent networks show that conformal normalization leads to the emergence of the hexagon grid patterns. Furthermore, we derive a new theoretical understanding that connects conformal normalization to the emergence of hexagon grid patterns in navigation tasks.

## 1 Introduction

The mammalian hippocampus formation encodes a "cognitive map" (Tolman, 1948; O'keefe & Nadel, 1979) of the animal's surrounding environment. In the 1970s, it was found that the rodent hippocampus contained place cells (O'Keefe & Dostrovsky, 1971), which typically fired at specific locations in the environment. Several decades later, another prominent type of neurons called grid cells (Hafting et al., 2005; Fyhn et al., 2008; Yartsev et al., 2011; Killian et al., 2012; Jacobs et al., 2013; Doeller et al., 2010) were discovered in the medial entorhinal cortex. Each grid cell fires at multiple locations that form a hexagonal periodic grid over the field (Fyhn et al., 2004; Hafting et al., 2005; Fuhs & Touretzky, 2006; Burak & Fiete, 2009; Sreenivasan & Fiete, 2011; Blair et al., 2007; Couey et al., 2013; de Almeida et al., 2009; Pastoll et al., 2013; Agmon & Burak, 2020). Grid cells interact with place cells and are believed to be involved in path integration (Hafting et al., 2005; Fiete et al., 2008; McNaughton et al., 2006; Gil et al., 2018; Ridler et al., 2019; Horner et al., 2016), which calculates the agent's self-position by accumulating its self-motion, allowing the agent to determine its location even when navigating in darkness. Thus, grid cells are often considered to form an internal GPS system in the brain (Moser & Moser, 2016). While grid cells were mostly studied in the spatial domain, it was proposed that grid-like response may also exist in non-spatial and more abstract cognitive spaces (Constantinescu et al., 2016; Bellmund et al., 2018).

Various computational models have been proposed to explain the striking firing properties of grid cells. Traditional approach designed hand-crafted continuous attractor neural networks (CANN) (Amit, 1992; Burak & Fiete, 2009; Couey et al., 2013; Pastoll et al., 2013; Agmon & Burak, 2020) and studied them by simulation. More recently two pioneering papers (Cueva & Wei, 2018; Banino et al., 2018) learned recurrent neural networks (RNNs) on path integration tasks and demonstrated that grid patterns emerge in the learned networks. These results have been further developed in (Gao et al., 2019; Sorscher et al., 2019; Cueva et al., 2020; Gao et al., 2021; Whittington et al., 2021; Dorrell et al., 2022; Xu et al., 2022). In addition to RNN models, principal component analysis (PCA)-based basis expansion models (Dordek et al., 2016; Sorscher et al., 2019; Stachenfeld et al., 2017) with non-negativity constraints have been proposed to model the interaction between grid cells and place cells.

While prior work has shed much light on the grid cells, the mathematical principle and the computational mechanisms that underlie the emergence of hexagon grid patterns are still not well understood (Cueva & Wei, 2018; Sorscher et al., 2023; Gao et al., 2021; Nayebi et al., 2021; Schaeffer et al., 2022). The goal of this paper is to propose a simple and general mechanism in the recurrent neural network of grid cells that leads to the emergence of hexagon grid patterns of grid cells.

Specifically, the activities of the population of grid cells collectively form a vector in a high-dimensional neural space. This high-dimensional vector is a representation of the 2D self-position of the agent in the 2D physical space. Adopting terminology in the deep learning literature, we call this vector the position embedding (Vaswani et al., 2017). As the agent navigates in the environment, the position embedding is transformed by a recurrent neural network that takes the velocity of the agent as input. For the recurrent network, we propose a novel conformal normalization mechanism that modulates the input velocity by the $\ell_2$-norm of the directional derivative of the transformation defined by the recurrent network. Under conformal normalization, the local displacement of the position embedding in the high-dimensional neural space is proportional to the local displacement of the agent in the 2D physical space, regardless of the direction of the input self-velocity. As a consequence, the 2D Euclidean space is embedded conformally as a 2D manifold in the neural space, and this 2D manifold forms an internal 2D coordinate system of the 2D physical environment, thus mathematically realizing the notion that grid cells form an internal GPS system (Moser & Moser, 2016).

We then numerically examine two minimally simple models of the recurrent network. One is a linear model that models the movement of the position embedding on the 2D manifold. The other is a non-linear model that additionally also constrains the 2D manifold as the fixed points of the non-linear transformation when the input velocity is zero. Our numerical experiments show that our proposed conformal normalization leads to the hexagon grid patterns in both models. We also provide a new theoretical understanding that connects the conformal normalization to the emergence of the hexagon grid patterns in the general setting.

Our work provides a novel mechanism that leads to the hexagon grid patterns of grid cells. Our linear and non-linear models based on the proposed mechanism may serve as useful building blocks for future modeling of grid cells and place cells. In summary, our contributions are as follows. (1) We propose a simple and general conformal normalization mechanism for the recurrent neural network of grid cells that leads to hexagon grid patterns observed in grid cells. (2) Our numerical experiments on both linear and non-linear models demonstrate that conformal normalization leads to the emergence of hexagon grid patterns. (3) We provide a theoretical understanding that connects our conformal normalization mechanism to hexagon grid patterns in the general setting.

## 2 BACKGROUND: POSITION EMBEDDING AND TRANSFORMATION

This section provides the background of position embedding and recurrent transformation. See (Gao et al., 2021) for more details.

### 2.1 POSITION EMBEDDING

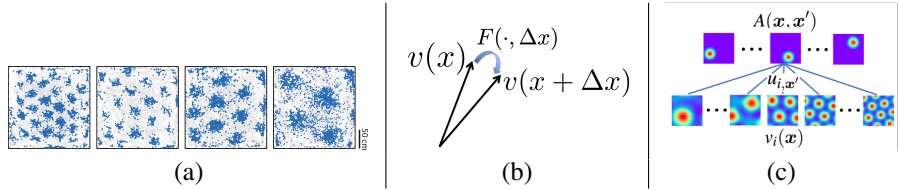

(a)      (b)      (c)

Figure 1: (a) Recorded response maps of 4 different grid cells (from Moser et al. (2014)). (b) The self-position $\boldsymbol{x} = (x_1, x_2)$ in 2D physical space is represented by a vector $\boldsymbol{v}(\boldsymbol{x})$ in the $d$-dimensional neural space. When the agent moves by $\Delta \boldsymbol{x}$, the vector is transformed to $\boldsymbol{v}(\boldsymbol{x} + \Delta \boldsymbol{x}) = F(\boldsymbol{v}(\boldsymbol{x}), \Delta \boldsymbol{x})$. (c) Illustration of basis expansion model $A(\boldsymbol{x} \mid \boldsymbol{x}') = \sum_{i=1}^{d} u_{i, \boldsymbol{x}'} v_i(\boldsymbol{x})$, where $v_i(\boldsymbol{x})$ is the response map of $i$-th grid cell, shown at the bottom. $A(\boldsymbol{x} \mid \boldsymbol{x}')$ is the response map of place cell associated with $\boldsymbol{x}'$, shown at the top. $u_{i, \boldsymbol{x}'}$ is the connection weight (from Gao et al. (2021)).

When the agent is at the self-position $x = (x_1, x_2)$ within a 2D domain in $\mathbb{R}^2$, the activities of the population of grid cells form a vector $v(x) = (v_i(x), i = 1, ..., d)$, where $v_i(x)$ is the activity of the $i$-th grid cell at position $x$. See Figure 1(b). The dimensionality $d$ is the number of grid cells. We call the space of $v$ the neural space, and we embed the 2D $x$ as a vector $v(x)$ in the $d$-dimensional neural space. We call this vector the position embedding by adopting the commonly used terminology in deep learning (Vaswani et al., 2017). We normalize $\|v(x)\| = 1$, so that $v(x)$ is on the $(d-1)$-sphere. $\|v(x)\|^2$ can be interpreted as the total energy of the neurons in $v(x)$.

For each grid cell $i$, $v_i(x)$, as a function of $x$, represents the response map of grid cell $i$. The intriguing observation in neuroscience is that the response map exhibits a periodic hexagonal grid pattern, with different grid cells having varying scales, orientations, and spatial shifts (phases). Figure 1(a) displays the response maps of four different grid cells.

## 2.2 RECURRENT TRANSFORMATION

At self-position $x = (x_1, x_2)$, if the agent makes a movement $\Delta x = (\Delta x_1, \Delta x_2)$, then it moves to $x + \Delta x$. Correspondingly, the vector $v(x)$ is transformed to $v(x + \Delta x)$. The general form of the transformation can be formulated as:

$$v(x + \Delta x) = F(v(x), \Delta x), \tag{1}$$

where $F$ can be parametrized by a recurrent neural network (RNN), and the recurrent transformation $F$ takes $\Delta x$ as an input. See Figure 1(b). We may call $\Delta x$ the input velocity if we assume a unit time period for the movement. We call (1) the recurrent transformation model.

The input velocity $\Delta x$ can also be represented as $(\Delta r, \theta)$ in polar coordinates, where $\Delta r$ is the displacement along the direction $\theta \in [0, 2\pi]$, so that $\Delta x = (\Delta x_1 = \Delta r \cos \theta, \Delta x_2 = \Delta r \sin \theta)$. The transformation model then becomes

$$v(x + \Delta x) = F(v(x), \Delta r, \theta), \tag{2}$$

where we continue to use $F(\cdot)$ for the recurrent transformation (slightly overloading the notation).

## 2.3 PLACE CELLS

The vector $v(x)$ serves to inform the agent of its adjacency to different positions, via a linear read-out mechanism:

$$A(x \mid x') = \langle v(x), u(x') \rangle = \sum_{i=1}^{d} u_i(x') v_i(x), \tag{3}$$

where $A(x \mid x')$, as a function of $x$, can be considered as the response map of the place cell associated with the place $x'$. In open field, the measured response map $A(x \mid x')$ can be well approximated by a Gaussian adjacency kernel $A(x \mid x') = \exp(-\|x - x'\|^2 / 2\sigma^2)$ for a certain scale parameter $\sigma$. $u(x') = (u_i(x'), i = 1, ..., d)$ is a $d$-dimensional read-out vector and can be regarded as the connection weight from grid cell $i$ to the place cell associated with $x'$. The right-hand side of Equation (3) implies that the response maps of grid cells $v_i(x)$ may serve as basis functions to expand the response map $A(x \mid x')$ of place cell associated with $x'$ (Figure 1(c)). We call (3) the basis expansion model.

**Path integration**. The above recurrent transformation model (1) and the basis expansion model (3) enable the agent to navigate. Suppose the agent starts from $x_0$, with vector representation $v_0 = v(x_0)$. If the agent makes a sequence of moves $(\Delta x_t, t = 1, ..., T)$, then the vector $v$ is updated by $v_t = F(v_{t-1}, \Delta x_t)$. At time $t$, the self-position of the agent can be decoded by $\hat{x} = \arg\max_{x'} \langle v_t, u(x') \rangle$, i.e., the place $x'$ that is the most adjacent to the self-position represented by $v_t$. This enables the agent to infer and keep track of its position based on its self-motion even in darkness.

## 3 CONFORMAL NORMALIZATION

This section presents our conformal normalization mechanism for the recurrent transformation.

### 3.1 Normalization in neuroscience and deep learning

Divisive normalization is a canonical operation widely observed in the cortex and has been extensively used in previous models in computational neuroscience (Geisler & Albrecht, 1992; Carandini & Heeger, 2012; Heeger, 1992; Schwartz & Simoncelli, 2001), which may emerge due to the recurrent computations between the excitatory and inhibitory neurons (Rubin et al., 2015; Niell, 2015). In deep learning models, batch normalization (Ioffe & Szegedy, 2015; Ioffe, 2017), layer normalization (Ba et al., 2016) and group normalization (Wu & He, 2018) are ubiquitous and indispensable.

### 3.2 Definition

Consider the general recurrent transformation $v(x + \Delta x) = F(v(x), \Delta r, \theta)$, where $\Delta x = (\Delta x_1 = \Delta r \cos \theta, \Delta x_2 = \Delta r \sin \theta)$, $\theta$ is the heading direction, and $\Delta r$ is the displacement.

**Definition 1** *The directional derivative of $F$ at $(v, \theta)$ is defined as*

$$f(v, \theta) = \frac{\partial}{\partial a} F(v, a, \theta) \mid_{a=0} . \tag{4}$$

With the above definition, the first order Taylor expansion at $\Delta r = 0$ gives us

$$v(x + \Delta x) = v(x) + f(v(x), \theta) \Delta r + o(\Delta r). \tag{5}$$

We define conformal normalization of the recurrent transformation so that the displacement of the position vector in the neural space is proportional to the displacement of the agent in the 2D physical space, regardless of the direction of the movement. This can be achieved by modulating the input velocity by the norm of the directional derivative of the transformation.

**Definition 2** *The conformal normalization of $\Delta r$ at $v(x)$ is defined as*

$$\overline{\Delta r} = \frac{s \Delta r}{\|f(v(x), \theta)\|}, \tag{6}$$

*where $\| \cdot \|$ is the $\ell_2$ norm, $s$ is either a learnable parameter or the average of $\|f(v(x), \theta)\|$ over the directions $\theta$.*

*The conformal normalization of the original recurrent transformation $F(v(x), \Delta r, \theta)$ is defined as $F(v(x), \overline{\Delta r}, \theta)$, so that after conformal normalization, the transformation is changed to*

$$v(x + \Delta x) = F(v(x), \overline{\Delta r}, \theta). \tag{7}$$

### 3.3 Conformal 2D manifold

**Proposition 1** *With conformal normalization (6) and (7), we have*

$$\|v(x + \Delta x) - v(x)\| = s\|\Delta x\| + o(\|\Delta x\|). \tag{8}$$

*(8) is called conformal isometry.*

The proof is straightforward. For the transformation (7), the first order Taylor expansion gives

$$v(x + \Delta x) = v(x) + f(v(x), \theta) \overline{\Delta r} + o(\Delta r) = v(x) + s\overline{f}(v(x), \theta) \Delta r + o(\Delta r), \tag{9}$$

where $\overline{f}(v, \theta) = f(v, \theta) / \|f(v, \theta)\|$ is a unit vector with $\|\overline{f}(v, \theta)\| = 1$, which leads to (8).

Conformal isometry (8) means that as the agent moves by $\|\Delta x\|$ in the 2D physical space, the position embedding $v$ moves by $s\|\Delta x\|$ in the $d$-dimensional neural space, regardless of the heading direction $\theta$. Conformal isometry leads to conformal embedding, i.e., a local coordinate system around $x$ (e.g., a polar coordinate system) is mapped to a local coordinate system around $v(x)$ without distortion of shape except for a scaling factor $s$.

Suppose the agent navigates within a 2D domain $\mathbb{D}$, e.g., a square, and if $s$ is a constant over $x$, then the 2D domain $\mathbb{D}$ is embedded as a 2D manifold $\mathbb{M} = (v(x), x \in \mathbb{D})$ in the neural space. This 2D

manifold $\mathbb{M}$ is conformal to the 2D physical domain $\mathbb{D}$. We may imagine $\mathbb{D}$ as a piece of flat paper. We can fold or bend it into $\mathbb{M}$ without distortion by stretching except for a global scaling factor $s$. Then $\mathbb{M}$ forms a 2D coordinate system of the physical domain without distortion except global scaling, e.g., magnification, so that the local distance between two position embeddings informs the agent of the physical distance between the two positions. The movement of $v$ on $\mathbb{M}$ can be realized by the recurrent transformation $F$. Thus $(\mathbb{M}, F)$ becomes a mathematical realization of an internal GPS system (Moser & Moser, 2016).

In fact, the positions $x$ and $x'$ in our model do not need to be actual 2D coordinates and there is no need to assume an *a priori* 2D coordinate system. $x$ and $x'$ can be discretized and indexed by discrete indices. Our model only needs to know the heading direction $\theta$ and self-displacement $\Delta r$ that connect different nearby positions. The learned $v$ associated with position $x$ (which may be just an index) will form the 2D coordinate of $x$. That is, our model learns to place the positions on a conform 2D coordinate system.

Is it too wasteful to use a high-dimensional $v$ to represent 2D coordinates? The answer is no. $v$ can inform the agent of its adjacency to any position $x'$ via a linear read-out vector $u(x')$ (i.e., linear probing), even though adjacency $A(x|x')$ is highly non-linear in $x$ and $x'$. That is, $v(x)$ serves as linear basis functions to expand any non-linear value functions of $x$. This is related to the Peter-Weyl theory (Taylor, 2002) where group representation gives rise to linear basis functions.

### 3.4 LINEAR MODEL

We shall numerically study the following linear model:

$$v(x + \Delta x) = (I + B(\theta)\Delta r)v(x) = v(x) + B(\theta)v(x)\Delta r. \tag{10}$$

where the directional derivative is $f(v, \theta) = B(\theta)v(x)$. Thus the conformal normalization is

$$\overline{\Delta r} = \frac{s\Delta r}{\|B(\theta)v(x)\|}. \tag{11}$$

The conformal normalization of the linear model then becomes

$$v(x + \Delta x) = v(x) + s\frac{B(\theta)v(x)}{\|B(\theta)v(x)\|}\Delta r. \tag{12}$$

The above model is similar to the "add + layer norm" operations in the Transformer model (Vaswani et al., 2017).

### 3.5 NON-LINEAR MODEL

We shall also numerically study the following non-linear model:

$$v(x + \Delta x) = R(Wv(x) + B(\theta)v(x)\Delta r), \tag{13}$$

where $W$ is a learnable matrix, and $R()$ is element-wise non-linear rectification, such as Tanh and GeLU (Hendrycks & Gimpel, 2016). For this model, the directional derivative is

$$f(v, \theta) = R'(Wv) \odot B(\theta)v, \tag{14}$$

where $R'()$ is calculated element-wise, and $\odot$ is element-wise multiplication. The conformal normalization then follows (6) and (7).

While the linear model is defined for $v(x) \in \mathbb{M}$ on the manifold, the non-linear model further constrains $v(x) = R(Wv(x))$ for $v(x) \in \mathbb{M}$, where $\Delta r = 0$. If $R(Wv)$ is furthermore a contraction for $v$ that are off $\mathbb{M}$, then $\mathbb{M}$ consists of the attractors of $R(Wv)$ for $v$ around $\mathbb{M}$. The non-linear model (13) then becomes a continuous attractor neural network (CANN) (Amit, 1992; Burak & Fiete, 2009; Couey et al., 2013; Pastoll et al., 2013; Agmon & Burak, 2020).

See Appendix A.1.1 for eigen analysis.

### 3.6 MULTIPLE BLOCKS AND MULTI-SCALE COORDINATE SYSTEMS

The grid cells form multiple modules or blocks (Barry et al., 2007; Stensola et al., 2012), and the response maps of grid cells within each module share the same scale. We thus assume that $B(\theta)$ is block diagonal, i.e., $v(x)$ consists of sub-vectors, and each sub-vector is operated on by a sub-matrix on the diagonal of $B(\theta)$. In the non-linear model, we assume $W$ is a full matrix. For each sub-vector, we normalize its $\ell_2$ norm to be the same constant. Each sub-vector is on a 2D manifold, which serves as a coordinate system at a particular scale. For multiple modules, we have coordinate systems of multiple scales or resolutions. The notion of local distance also changes with scale.

### 3.7 LEARNING

To learn the system, we discretize $x \in \mathbb{D}$ and $\theta \in [0, 2\pi]$. The input consists of place cell adjacency kernels $A(x \mid x')$. The output consists of $(v(x), u(x'), B(\theta))$ for the linear model, and additionally $W$ for the non-linear model. The loss terms are:

$$L_0 = \mathbb{E}_{x,x'}[(A(x|x') - \langle v(x), u(x')\rangle)^2], \tag{15}$$

$$L_1 = \mathbb{E}_{x,\Delta x}[\|v(x + \Delta x) - F(v(x), \Delta x)\|^2], \tag{16}$$

where $F()$ is the transformation after conformal normalization. The expectations can be approximated by Monte Carlo averages of random samples of $x$, $x'$, and $\Delta x$ within their ranges. $L_0$ is for the basis expansion of place cell kernels. We assume $u(x') \geq 0$ because the connections from grid cells to place cells are excitatory (Zhang et al., 2013; Rowland et al., 2018). $L_1$ is for one-step transformation in path integration.

In the numerical experiments, we jointly learn the position embedding $v(x)$, read-out weights $u(x')$, and transformation model $F()$ by minimizing the total loss: $L = L_0 + \lambda_1 L_1$. A special case of $L_1$ with $\Delta x = 0$ enforces the fixed point condition.

## 4 THEORETICAL UNDERSTANDING

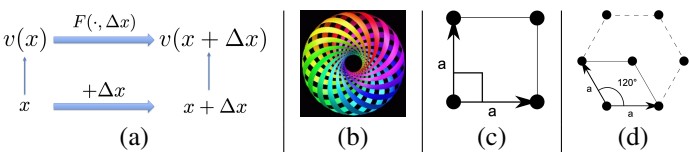

Figure 2: (a) $(F(\cdot, \Delta x), \forall \Delta x \in \mathbb{R}^2)$ is a group of transformations, and this transformation group is a representation of the 2D additive Euclidean group $(\mathbb{R}^2, +)$. (b) A 2D torus embedded in 3D space (from Banchoff (1968)). (c) Square lattice. (d) Hexagon lattice.

In this section, we seek to connect our conformal normalization to the hexagon grid pattern in the general setting. In the following, Step 1 follows Gao et al. (2021), and Step 2 substantially expands Xu et al. (2022). We add Step 3 to justify the hexagon lattice.

**Step 1: Abelian Lie group**. The group of transformation $(F(\cdot, \Delta x), \forall \Delta x)$ acting on the manifold $(v(x), \forall x)$ form a representation of the 2D additive Euclidean group $(\mathbb{R}^2, +)$, i.e., $F(v(x), \Delta x_1 + \Delta x_2) = F(F(v(x), \Delta x_1), \Delta x_2) = F(F(v(x), \Delta x_2), \Delta x_1)$, $\forall x, \Delta x_1, \Delta x_2$, and $F(v(x), 0) = v(x)$, $\forall x$. See Figure 2(a) for an illustration. Since $(\mathbb{R}^2, +)$ is an abelian Lie group, $(F(\cdot, \Delta x), \forall \Delta x)$ is also an abelian Lie group.

**Step 2: Torus topology**. Because the elements of $v(x)$ are neuron firing rates, they are bounded. Thus the manifold $(v(x), \forall x)$ is compact, and $(F(\cdot, \Delta x), \forall \Delta x)$ is a compact group. It is also connected because the 2D domain is connected. According to a classical theorem in Lie group theory (Dwyer & Wilkerson, 1998), a compact and connected abelian Lie group has a topology of a torus, i.e., $\mathbb{S}_1^r$, where each $\mathbb{S}_1$ is topologically a circle, and $r$ is the rank or dimensionality of the torus.

There are multiple blocks (or modules) in $v(x)$, each of which is operated separately by a block of the block-diagonal $B(\theta)$. We may assume each block (or module) has the minimal rank 2 (rank 1 can be considered a degenerate special case, see below). Otherwise, we can continue to divide the

block into sub-blocks, each of which operates separately. For notation simplicity, we continue to use $F(\cdot, \Delta\boldsymbol{x})$ and $\boldsymbol{v}(\boldsymbol{x})$ to denote the transformation and position embedding of a single block. If the torus formed by $(F(\cdot, \Delta\boldsymbol{x}), \forall\Delta\boldsymbol{x})$ is 2D, then its topology is $\mathbb{S}_1 \times \mathbb{S}_1$, where each $\mathbb{S}_1$ is a circle. Thus we can find two 2D vectors $\Delta\boldsymbol{x}_1$ and $\Delta\boldsymbol{x}_2$, so that $F(\cdot, \Delta\boldsymbol{x}_1) = F(\cdot, \Delta\boldsymbol{x}_2) = F(\cdot, 0)$. As a result, $\boldsymbol{v}(\boldsymbol{x})$ is a 2D periodic function so that $\boldsymbol{v}(\boldsymbol{x} + k_1\Delta\boldsymbol{x}_1 + k_2\Delta\boldsymbol{x}_2) = \boldsymbol{v}(\boldsymbol{x})$ for arbitrary integers $k_1$ and $k_2$. We assume $\Delta\boldsymbol{x}_1$ and $\Delta\boldsymbol{x}_2$ are the shortest vectors that characterize the above 2D periodicity. According to the theory of 2D Bravais lattice (Ashcroft et al., 1976) (see Appendix A.1.3 for details), any 2D periodic lattice can be defined by two primitive vectors $(\Delta\boldsymbol{x}_1, \Delta\boldsymbol{x}_2)$ (rank 1 degenerate case corresponds to one primitive vector being 0). The torus topology is supported by neuroscience data (Gardner et al., 2022). A 2D torus is commonly visualized as a donut shape in 3D space, as shown in Figure 2(b). But it can be more naturally imagined as a 2D rectangle with periodic boundary conditions.

If the scaling factor $s$ is constant over different positions, then as the position $\boldsymbol{x}$ of the agent moves from 0 to $\Delta\boldsymbol{x}_1$ in the 2D space, $\boldsymbol{v}(\boldsymbol{x})$ traces a perfect circle of circumference $s\|\Delta\boldsymbol{x}_1\|$ in the neural space due to conformal isometry, i.e., the geometry of the trajectory of $\boldsymbol{v}(\boldsymbol{x})$ is a perfect circle up to bending or folding but without distortion by stretching. The same with movement from 0 to $\Delta\boldsymbol{x}_2$. Since we normalize $\|\boldsymbol{v}(\boldsymbol{x})\|$ to be a constant, the two circles have the same radius and thus they also have the same circumferences, hence we have $\|\Delta\boldsymbol{x}_1\| = \|\Delta\boldsymbol{x}_2\|$ (which also implies that rank 1 degenerate case is forbidden by conformal normalization). According to Bravais lattice theory (Ashcroft et al., 1976), the periodic lattice with two equal-length primitive vectors can only be square or hexagon, as illustrated by Figure 2(c) and (d).

**Step 3: Fourier analysis**. The Fourier transform of a 2D period function $f(\boldsymbol{x})$ can be written as a linear superposition of Fourier components $f(\boldsymbol{x}) = \sum_k \hat{f}(\omega_k)e^{i\langle\omega_k,\boldsymbol{x}\rangle}$, where $\omega_k = k_1\boldsymbol{a}_1 + k_2\boldsymbol{a}_2$, $k = (k_1, k_2)$ are two integers, and $(\boldsymbol{a}_1, \boldsymbol{a}_2)$ are primitive vectors in the reciprocal space. For square or hexagon lattice with $\|\Delta\boldsymbol{x}_1\| = \|\Delta\boldsymbol{x}_2\| = \rho$, we have $\|\boldsymbol{a}_1\| = \|\boldsymbol{a}_2\| = 2\pi/\rho$, and the lattice in the reciprocal space remains to be square or hexagon respectively. For a 2D Gaussian adjacent kernel centered at origin, $A(\boldsymbol{x}) = \frac{1}{2\pi\sigma^2}\exp(-\|\boldsymbol{x}\|^2/2\sigma^2)$, its 2D Fourier transform is $\hat{A}(\omega) = \exp(-\sigma^2\|\omega\|^2/2)$, which goes to zero as $\|\omega\| \to \infty$. Therefore we only need to consider frequency components $\Omega = \{\omega : \|\omega\| \leq D\}$ for a big enough $D$. For each $\boldsymbol{x}$, let $\boldsymbol{e}(\boldsymbol{x}) = (e^{i\langle\omega_k,\boldsymbol{x}\rangle}, \omega_k \in \Omega)$ be the column vector formed by the Fourier components within $\Omega$. Let $\boldsymbol{v}(\boldsymbol{x}) = \boldsymbol{M}\boldsymbol{e}(\boldsymbol{x})$ for a matrix $\boldsymbol{M}$. Let us assume the dimensionality of $\boldsymbol{v}(\boldsymbol{x})$ is no less than the dimensionality of $\boldsymbol{e}(\boldsymbol{x})$. Then the least square regression on $\boldsymbol{v}(\boldsymbol{x})$ amounts to the least squares regression on $\boldsymbol{e}(\boldsymbol{x})$. The hexagon lattice packs more Fourier components into $\Omega$ than the square lattice with the same $\|\boldsymbol{a}_1\| = \|\boldsymbol{a}_2\| = 2\pi/\rho$. All these discrete Fourier components are orthogonal to each other. Thus the hexagon $\boldsymbol{v}(\boldsymbol{x})$ provides a better least squares fit to the kernel function $A(\boldsymbol{x})$ in the basis expansion. Different blocks have different scales of $\|\boldsymbol{a}_1\| = \|\boldsymbol{a}_2\|$ as well as different orientations to pave the whole frequency domain. Our results expand upon previous theoretical arguments using optimal packing density to justify the optimality of hexagonal grids (Wei et al., 2015; Mathis et al., 2015). See Appendix A.1 for more on theoretical understanding.

## 5 EXPERIMENTS

We conducted numerical experiments to train both linear and non-linear models. Within these experiments, we use a square open area measuring $1\text{m} \times 1\text{m}$ that was subdivided into $40 \times 40$ spatial bins. The dimensions of $\boldsymbol{v}(\boldsymbol{x})$, representing the total number of grid cells, are 360 for the linear model and 192 for the non-linear model. For both models, we partition $\boldsymbol{v}(\boldsymbol{x})$ into multiple blocks with block size 24.

For the response map of the place cell associated with $\boldsymbol{x}'$, we use the Gaussian adjacency kernel with $A(\boldsymbol{x}|\boldsymbol{x}') = \exp(-\|\boldsymbol{x} - \boldsymbol{x}'\|^2/(2\sigma^2))$, where $\sigma = 0.07$. For transformation, the one-step displacement $\Delta r$ is set to be smaller than 3 grids. The scaling factor $s$ is taken to be the average of $\|f(\boldsymbol{v}(\boldsymbol{x}), \theta)\|$ over $\theta$. $s$ can also be a learnable parameter, and Appendix A.2.1 contains result with learnable $s$.

### 5.1 HEXAGON PATTERNS

Figures 3 and 4 show the learned firing patterns of $\boldsymbol{v}(\boldsymbol{x}) = (v_i(\boldsymbol{x}), i = 1, ..., d)$ over the $40 \times 40$ lattice of $\boldsymbol{x}$ for linear and non-linear models. Each image represents the response map for a grid cell, with every row displaying the units learned within the same module. The emergence of hexagonal

patterns in these activity patterns is evident. Within each module, scales, and orientations remain consistent, but they exhibit different phases or spatial shifts. Our findings highlighted the essential role of conformal normalization; in its absence, the response maps displayed non-hexagon or stripe-like patterns. Ablation results can be found in Appendix A.2.2. To show generality, we also provide results for both models with different block sizes.

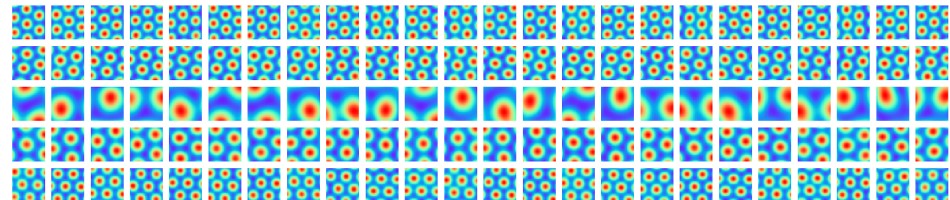

Figure 3: In the linear model, hexagon grid firing patterns are observed in the learned $v(x)$. Each row displays the firing patterns of all the cells within a single module, with each module comprising 24 cells. The units illustrate the neuron activity throughout the entire 2D square environment. The figure presents patterns from five randomly chosen modules.

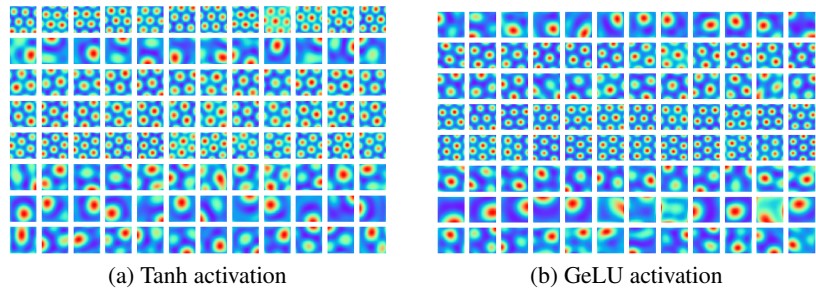

(a) Tanh activation                    (b) GeLU activation

Figure 4: Results of the non-linear models. We randomly chose 8 modules and showed the firing patterns with different rectification functions.

To evaluate how closely the learned patterns align with regular hexagonal grids, we report the gridness scores in Table 1. These scores are derived from grid cell literature (Langston et al., 2010; Sargolini et al., 2006). We also present the percentage of valid grid cells that meet the criteria of a gridness score greater than 0.37.

Table 1: Gridness scores and valid rates of grid cells of learned models.

| Model | Gridness score | % of grid cells |
|---|---|---|
| Banino et al. (2018) | 0.18 | 25.2 |
| Sorscher et al. (2019) | 0.48 | 56.1 |
| Gao et al. (2021) | 0.90 | 73.1 |
| Ours (Linear) | 0.86 | 82.5 |
| Ours (Non-linear) | 0.87 | 87.6 |

## 5.2 PATH INTEGRATION

We assess the ability of the learned model to execute accurate path integration in two different scenarios. First of all, for path integration with re-encoding, we decode $v \to \hat{x}$ to the 2D physical space via $\hat{x} = \arg\max_{x'} \langle v, u(x') \rangle$, subsequently encode $v \leftarrow v(\hat{x})$ back to the neuron space intermittently. This approach aids in rectifying the errors accumulated in the neural space throughout the transformation. Conversely, in scenarios excluding re-encoding, the transformation is applied exclusively using the neuron vector $v$. In the left panel of Figure 5, the model adeptly handles path integration up to 30 steps (short distance) without the need for re-encoding. For path integration with longer distances, we evaluate the learned model for 100 steps over 300 trajectories. As shown in the

right panel of Figure 5, with re-encoding, the path integration error for the last step is 0.003, while the average error over the 100-step trajectory is 0.002. Without re-encoding, the error is relatively larger, where the average error over the entire trajectory is approximately 0.024, and it reaches 0.037 for the last step.

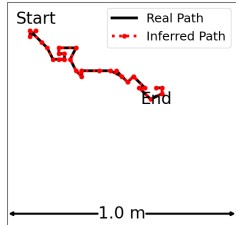 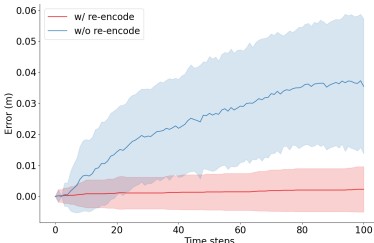

Figure 5: Results for path integration. *Left*: path integration for 30 steps without re-encoding. The black line represents the real trajectory and the red one is the inferred trajectory by the learned model. *Right*: results for long distance (100-step) path integration error with and without re-encoding over time by the non-linear model.

## 6    LIMITATIONS

Our work focuses on the study of grid cells. We assume that place cells are modeled by Gaussian adjacency kernels in the open environment. We do not seek to model place cells beyond that. On the other hand, our model of grid cells can potentially be integrated into a more sophisticated model of place cells. In our study of grid cells, we focus on general transformation and a simple and general conformal normalization mechanism. We study the linear model and non-linear model as concrete machine learning models due to their simplicity. It is possible that the biological grid cell system employs similar normalization scheme, but it is difficult to test this hypothesis against available neuroscience data.

## 7    RELATED WORK

In computational neuroscience, hand-crafted continuous attractor neural networks (CANN) (Amit, 1992; Burak & Fiete, 2009; Couey et al., 2013; Pastoll et al., 2013; Agmon & Burak, 2020) were designed for path integration. In machine learning, the pioneering papers (Cueva & Wei, 2018; Banino et al., 2018) learned RNNs for path integration. However, RNNs do not always learn hexagon grid patterns. PAC-based basis expansion models (Dordek et al., 2016; Stachenfeld et al., 2017) and some theoretical accounts based on learned RNNs (Sorscher et al., 2023) rely on non-negativity assumption and the difference of Gaussian kernels for the place cells to explain the hexagon grid pattern. (Dorrell et al., 2022) proposes a loss function to learn grid cells.

Our work is based on (Gao et al., 2021; Xu et al., 2022), where the conformal isometry is constrained by an extra loss term that is rather unnatural. In contrast, in our work, the conformal isometry is built into the recurrent network *intrinsically* via a simple and general normalization mechanism, so that there is no need for extra loss term. While (Gao et al., 2021) focuses on the linear model in numerical experiments, our paper studies the non-linear model extensively. Our paper also provides a deeper and more comprehensive theoretical understanding.

## 8    CONCLUSION

Divisive normalization has been extensively studied in neuroscience and is ubiquitous in modern deep neural networks. This paper proposes a conformal normalization mechanism for recurrent neural networks of grid cells, and shows that the conformal normalization leads to the emergence of hexagon grid patterns. The proposed normalization mechanism is simple and general, and it leads to a conform embedding of the 2D Euclidean space in the high-dimensional neural space, formalizing the notion that the grid cells collectively form an internal GPS system.

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

# A  APPENDIX

## A.1  MORE THEORETICAL UNDERSTANDING

### A.1.1  EIGEN ANALYSIS OF TRANSFORMATION

For the general transformation, $v(x + \Delta x) = F(v(x), \Delta x)$, we have

$$v(x) = F(v(x), 0), \tag{17}$$
$$v(x + \Delta x) = F(v(x + \Delta x), 0), \tag{18}$$

thus

$$\Delta v = v(x + \Delta x) - v(x) = F'_v(v(x))\Delta v + o(\|\Delta v\|), \tag{19}$$

where

$$F'_v(v) = \frac{\partial}{\partial \Delta} F(v + \Delta, 0) \mid_{\Delta = 0}. \tag{20}$$

Thus $\Delta v$ is in the 2D eigen subspace of $F'_v(v(x))$ with eigenvalue 1.

At the same time,

$$v(x + \Delta x) = F(v(x), \Delta x) = v(x) + F'_{\Delta x}(v(x))\Delta x, \tag{21}$$

where

$$F'_{\Delta x}(v) = \frac{\partial}{\partial \Delta x} F(v, \Delta x) \mid_{\Delta x = 0}. \tag{22}$$

Thus

$$\Delta v = F'_{\Delta x}(v(x))\Delta x, \tag{23}$$

that is, the two columns of $F'_{\Delta x}(v(x))$ are the two vectors that span the eigen subspace of $F'_v(v(x))$ with eigenvalue 1. If we further assume conformal embedding which can be enforced by conformal normalization, then the two column vectors of $F'_{\Delta x}(v(x))$ are orthogonal and of equal length, so that $\Delta v$ is conformal to $\Delta x$.

The above analysis is about $v(x)$ on the manifold. We want the remaining eigenvalues of $F'_v(v(x))$ to be less than 1, so that, off the manifold, $F(v, 0)$ will bring $v$ closer to the manifold, i.e., the manifold consists of attractor points of $F$, and $F$ is an attractor network.

### A.1.2 PERMUTATION GROUP

The learned response maps of the grid cells in the same module are shifted versions of each other, i.e., there is a discrete set of displacements $\{\Delta x\}$, such as for each $\Delta x$ in this set, we have $v_i(x + \Delta x) = v_j(x)$, where $j = \sigma(i, \Delta x)$, and $\sigma$ is a mapping from $i$ to $j$ that depends on $\Delta x$. In other words, $\Delta x$ causes a permutation of the indices of the elements in $v(x)$, and $F(\cdot, \Delta x) \cong \sigma(\cdot, \Delta x)$, that is, the transformation group is equivalent to a subgroup of the permutation group. This is consistent with hand-designed CANN. A CANN places grid cells on a 2D "neuron sheet" with periodic boundary condition, i.e., a 2D torus, and lets the movement of the "bump" formed by the activities of grid cells mirror the movement of the agent in a conformal way, and the movement of the "bump" amounts to cyclic permutation of the neurons. Our model does not assume such an *a priori* 2D torus neuron sheet, and is much simpler and more generic.

### A.1.3 BACKGROUND ON BRAVAIS LATTICE

Named after Auguste Bravais (1811-1863), the theory of Bravais lattice was developed for the study of crystallography in solid state physics.

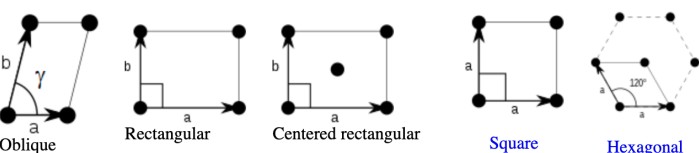

Figure 6: 2D periodic lattice is defined by two primitive vectors.

In 2D, a periodic lattice is defined by two primitive vectors $(\Delta x_1, \Delta x_2)$, and there are 5 different types of periodic lattices as shown in Figure 6. If $\|\Delta x_1\| = \|\Delta x_2\|$, then the two possible lattices are square lattice and hexagon lattice.

For Fourier analysis, we need to find the primitive vectors in the reciprocal space, $(a_1, a_2)$, via the relation: $\langle a_i, \Delta x_j \rangle = 2\pi \delta_{ij}$, where $\delta_{ij} = 1$ if $i = j$, and $\delta_{ij} = 0$ otherwise.

For a 2D periodic function $f(x)$ on a lattice whose primitive vectors are $(a_1, a_2)$ in the reciprocal space, define $\omega_k = k_1 a_1 + k_2 a_2$, where $k = (k_1, k_2)$ are a pair of integers (positive, negative, and zero), the Fourier expansion is $f(x) = \sum_k \hat{f}(\omega_k) e^{i\langle \omega_k, x \rangle}$.

See http://lampx.tugraz.at/~hadley/ss1/crystaldiffraction/fourier/ 2dBravais.php for more details. Figure 6 as well as Figure 2(c) and (d) are taken from the above webpage.

### A.2 MORE EXPERIMENT RESULTS

### A.2.1 LEARNED PATTERNS

In Figures 7 and 8, we show the autocorrelograms of the learned grid patterns from the linear and non-linear models.

We further tried with varying module sizes. Figure 9 visualizes the learned patterns when we fix the total number of grid cells but adjust the module size to 12 or 36. Importantly, the number or size of blocks doesn't impact the emergence of the hexagonal grid firing patterns.

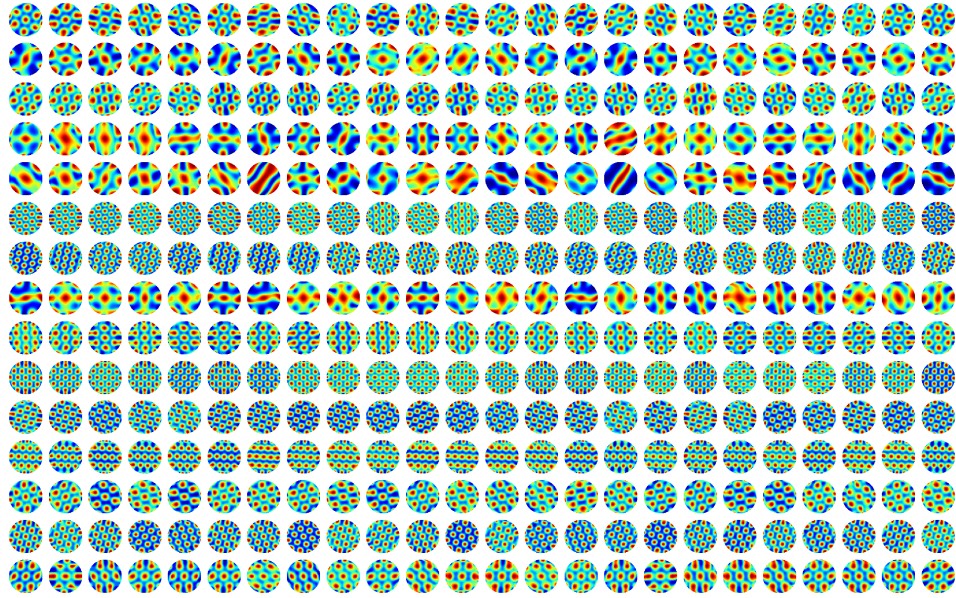

Figure 7: Autocorrelograms of the learned patterns for the linear model.

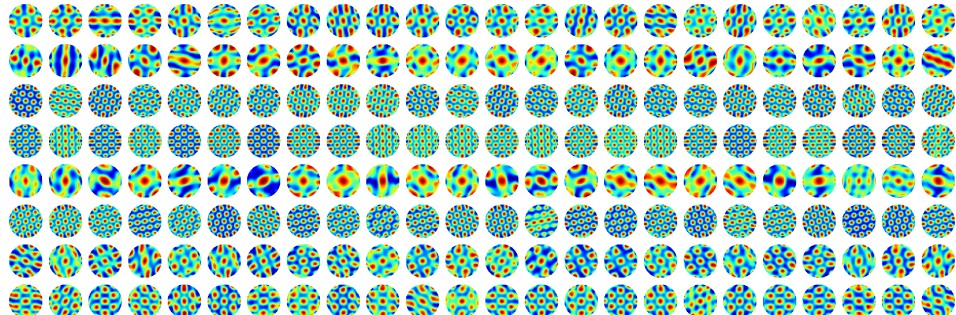

Figure 8: Autocorrelograms of the learned patterns for the non-linear model.

Furthermore, for the non-linear model, we experimented with different rectification functions, including GeLU. Our evaluations of the learned patterns yielded a gridness score of 0.87 and a ratio of grid cells at 78.65%. As depicted in Figure 10, hexagonal grid firing patterns can emerge using diverse activation functions.

Finally, for scaling factor $s$, we tried to learn it as a free parameter. In Figure 11, we show the learned hexagonal patterns for the linear model with 12 block size, which indicates that multi-scale grid patterns can be learned with or without learnable $s$.

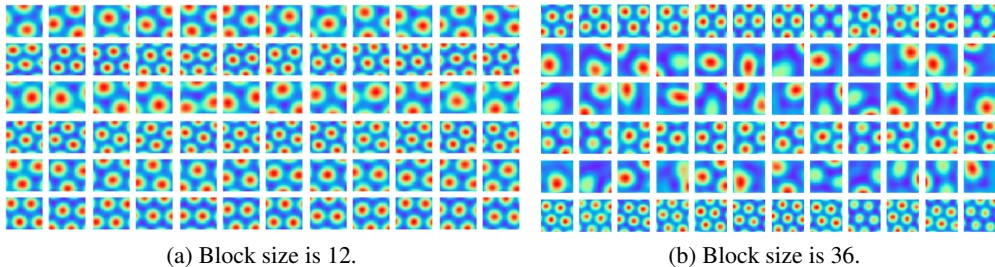

(a) Block size is 12.  (b) Block size is 36.

Figure 9: For the linear model, the learned patterns of $v(x)$ with 12 and 36 cells in each block. We randomly select 6 blocks for each model and show 12 cells of those blocks.

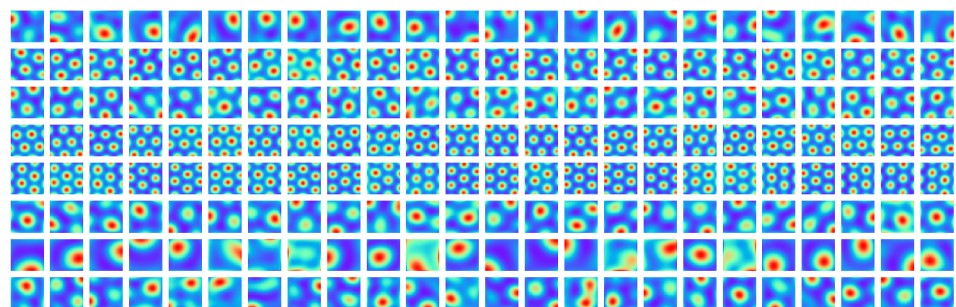

Figure 10: Firing patterns of the non-linear model with GeLU activation.

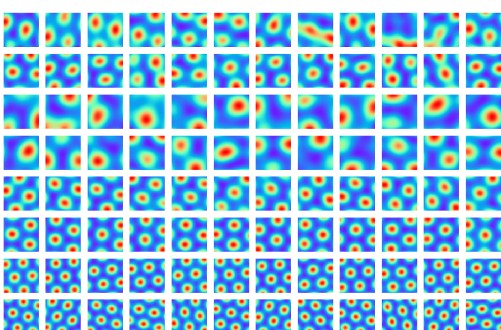

Figure 11: Learned patterns with learnable scaling factor *s*.

### A.2.2 ABLATION STUDIES

In this section, we show ablation results to investigate the empirical significance of certain components in our model for the emergence of hexagon grid patterns. First, the emergence of hexagon patterns is dependent on conformal normalization for both linear and non-linear models. Also, it is necessary for $B(\theta)$ to be a block-diagonal matrix in order to learn multi-scale hexagon firing patterns.

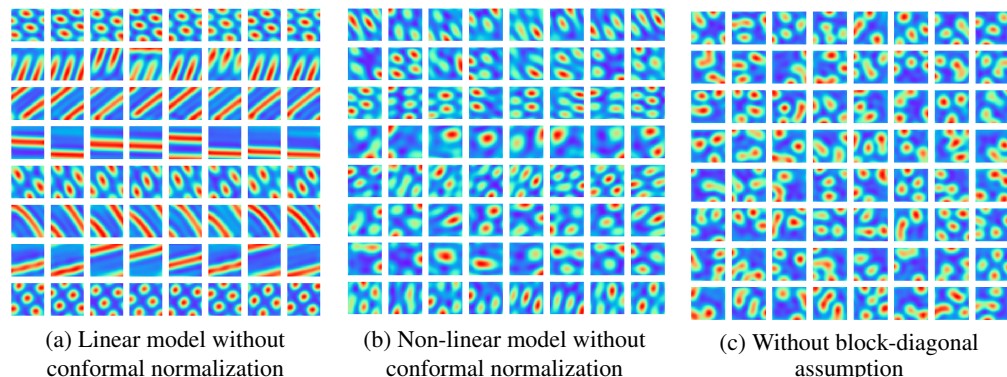

| (a) Linear model without conformal normalization | (b) Non-linear model without conformal normalization | (c) Without block-diagonal assumption |

Figure 12: Results of ablation on certain components of the model. (a) Learned patterns without conformal normalization in the linear model. (b) Learned patterns without conformal normalization in the non-linear model. (c) Learned patterns without the block-diagonal assumption for $B(\theta)$ in the linear model.

