# OpenReview forum: "Conformal Normalization in Recurrent Neural Network of Grid Cells"
_ICLR.cc/2024/Conference — ICLR 2024 Conference Withdrawn Submission_

### Official Review · Reviewer_rnXm · 2023-10-22

**Soundness:** 2 fair
**Presentation:** 2 fair
**Contribution:** 1 poor
**Rating:** 3
**Confidence:** 3

**Summary:**

Extend the work & analyses of Gao et al. 2021 and Xu et al. 2022 to propose a "conformal normalization" recurrent operation, then train linear & nonlinear "conformally normalized" recurrent neural networks that learn grid cells.

**Strengths:**

Overall, I think this paper is clearly written with a coherent story.

**Weaknesses:**

## Clarification of the Goals & Criteria for Success

> “The goal of this paper is to propose a simple and general mechanism in the recurrent neural network of grid cells that leads to the emergence of hexagon grid patterns of grid cells.”

I very much like papers that have clear, focused goals, so I compliment the authors on this. However, I wonder if the authors can improve this goal in two ways:

1. What is inadequate/incomplete/insufficient in previous models? As cited by the authors, Sorscher et al. 2019/2023, Gao et al. 2021 and Xu et al. 2022 all propose models that lead to hexagonal grid patterns in deep recurrent networks, as well as more classical Continuous Attractor Networks. The authors also mention Dorrell et al. 2023. Can the authors please sharpen the goal to clarify why these previous papers’ models are inadequate/incomplete/insufficient?

2. What are the criteria for success? Can we sharpen what constitutes a satisfactory (or better, an ideal) “simple and general mechanism” for describing grid cells?

## Incremental and Minor Work

I wasn’t clear on how this manuscript differs from Gao et al. 2021 and especially from Xu et al. 2022. Some immediately obvious similarities:

- Definition 2 (Conformal Normalization) here seems like some algebraic rearrangement of Gao et al. 2021’s Section 2.6.
- Proposition 1 (Conformal Isometry) is nearly identical to Gao et al’s Theorem 1.
- Eqn 10 matches Gao et al.’s Eqn 9.

It seems like this paper is a minor modification to Xu et al. 2022, where the sole contribution of this paper is the conformal normalization (Eqn 6), and I’m unsure whether such a contribution is too incremental and minor to merit a full contribution compared against Gao & Xu.

> Our work is based on (Gao et al., 2021; Xu et al., 2022), where the conformal isometry is constrained by an extra loss term that is rather unnatural. In contrast, in our work, the conformal isometry is built into the recurrent network intrinsically via a simple and general normalization mechanism, so that there is no need for extra loss term. While (Gao et al., 2021) focuses on the linear model in numerical experiments, our paper studies the non-linear model extensively.

This paragraph is quoted from Section 7, but I think it merits earlier mention and longer discussion.  Xu et al. 2022 make a similar claim that their work distinguishes itself from Gao et al. 2021: “Although Gao et al. (2021) studied general transformation model theoretically, they focused on a prototype model of linear recurrent network [...] In this paper, we study conformal isometry in the context of the non-linear recurrent model.”

## Realistic Modeling Assumptions?

When modeling grid cells, this paper makes 3 assumptions I'm uncomfortable with:

1. multiple grid modules are inserted by the researchers by making $B(\theta)$ block-diagonal (Section 3.6). If we want a good model of grid cells, it feels wrong to insert the number of grid modules in by hand.

2. The place cells have (a) uni-modal, (b) single-scale (c) Gaussian tuning curves and (d) are distributed uniformly over physical space (Section 2.3 and Section 5). Biologically, I don't think these are experimentally correct. My understanding is that place cells can have multiple fields, with multiple scales, and their tuning curves are not particularly Gaussian-like. I also understand that place cells' fields are not uniform in space but rather cluster together.

3. Physical space is discretized (assuming I understood Section 5 correctly). Many previous papers didn't make such assumptions, I believe, e.g., Banino et al., Sorscher et al., and I don't think physical space is discrete, at least at the scale of mammals moving through their daily lives. I also couldn't identify a reason why this choice was made, but perhaps I missed it?

## Weak Experimental Results

Overall, the experimental section feels weak to me:

- The environments considered (1m x 1m) are relatively small compared to previous works (2.2m x 2.2m)
- I don't know what effect(s) discretizing the spatial environment has. Maybe I'll better understand once I learn why the environment needed to be discretized
- There's no topological data analysis of the learned representations e.g., using the techniques used by Gardner et al. Nature 2022
- The error increase without re-encoding seems small. 3 cm after 100 time steps seems minor, especially since the position decoding error appears to be plateauing and since the position decoding error has some inherent error (Schaeffer et al. 2022 reported ~10 cm; I don't know whether you're using the same implementation)
- When $B(\theta)$ is block-diagonal, what prevents all the blocks from converging to approximately the same grid cell frequencies? Is there something that pushes each block to differentiate itself

**Questions:**

- I’m unclear on Eqn 7. Is Eqn 7 saying you replace Eqn 2 with the conformal normalization instead?

- In Section 5, I’m unclear on whether physical space was discretized. Is this the case? If so, why was that done, and is discretizing space necessary?

- Why are place cells necessary at all? I’m aware that previous papers, e.g., Banino et al 2018 included them, but I struggle to understand how they connect to the conformal normalization story.

- I think one of your citations is incorrect. "Dehong Xu, Ruiqi Gao, Wen-Hao Zhang, Xue-Xin Wei, and Ying Nian Wu. Conformal isometry
of lie group representation in recurrent network of grid cells. arXiv preprint arXiv:2210.02684, 2022." This paper was submitted to NeurReps 2022’s Proceedings Track and subsequently published in PMLR. Links to OpenReview https://openreview.net/pdf?id=FszPdSkvGjz and PMLR: https://proceedings.mlr.press/v197/

- In Equation 10, to confirm, $\delta r$ is a scalar? If so, then $B(\theta) \delta r$ is a matrix? I wasn’t quite clear while reading the text.

- When defining new symbols/terms (such as in Equations 4 or 7), can the authors please consider using a different notation than “=”? Even “:=” would be an improvement. This is to denote that the left hand side and right hand sides are definitionally equal.

- nit: Can the authors please change “and” to “or” in the sentence “R() is element-wise non-linear rectification, such as Tanh and GeLU (Hendrycks & Gimpel, 2016).”

---

> ### Author Response · Authors · 2023-11-22
>
> Thank you for your thorough and valuable feedback on our manuscript. We deeply value the time and expertise you've invested in reviewing our work.

---

### Official Review · Reviewer_AKJR · 2023-10-30

**Soundness:** 4 excellent
**Presentation:** 3 good
**Contribution:** 2 fair
**Rating:** 5
**Confidence:** 4

**Summary:**

In the submitted manuscript the authors introduce a model of grid cell formation that utilizes divisive normalization which, when combined with block-like organization of connectivity in grid modules, leads to the hallmark hexagonal organization. The work is technically sound and provides analytic justification for well-known receptive field organization, which may be of interest to those interested in explaining receptive fields found in spatial navigation tasks.

**Strengths:**

The work appears technically sound, with the analytic section providing an in-depth overview of the mapping from Euclidian space to activity space, and the numeric learning being described sufficiently to enable independent replication. The introduction of lateral inhibition is well justified by neuroscience literature, and as the authors note can circumvent the artificial introduction of auxiliary loss terms from previous publications while still leading to hexagonal representations.

**Weaknesses:**

While the paper is technically sound, the degree to which the manuscript overlaps with previous work, particularly Gao 2021 and Xu 2022, is too high to justify publication in a venue as impactful as ICLR. Many of the methods are directly comparable to the two works mentioned above, with the primary difference being the introduction of divisive normalization, and justification of hexagonal organization from frequency and packing arguments.

**Questions:**

-	Figure 5 presents results in path integration over a relatively small number of steps, and later alludes that the proposed model is a candidate for general self-localization. However, *many* models of path integration are capable of performing high fidelity integration over 10-100's of samples in the absence of sensory noise. Are there are any results (or can you generate them) to add to section 5.2 that show the proposed model outperforms previous approaches in the noiseless task, or run an additional experiment investigating robustness to noise?

---

> ### Author Response · Authors · 2023-11-22
>
> Thank you for your thorough and valuable feedback on our manuscript. We deeply value the time and expertise you've invested in reviewing our work.

---

### Official Review · Reviewer_DEe1 · 2023-10-31

**Soundness:** 2 fair
**Presentation:** 1 poor
**Contribution:** 1 poor
**Rating:** 3
**Confidence:** 4

**Summary:**

The paper addresses the critical question of self-positioning in the brain through grid cells and the vital role it plays. The paper proposes to explore the emergence of grid cells through the lens of conformal normalization in recurrent neural networks. The authors set up linear and nonlinear RNN models and aim to show that when considering a "conformal" normalization of the neural activities as a function of displacement, the model leads to the emergence of grid cells, unlike when those constraints are not fulfilled.

The authors propose some numerical experiments on path integration and visualization of the learned cells for different models with and without normalization.

**Strengths:**

The paper addresses an important question and might bring some insights.

**Weaknesses:**

The paper is not written rigorously, and notations are not presented clearly. The notations for polar coordinates are not conventional, and the multiple normalization steps to present the first-order approximation of the neural activity as a function of motion are poorly given.

I would suggest the authors start from the end, think about what is essential, and not over-emphasize the theory when they consider 2D motion. The heaviness of the theory is not crucial when considering relatively simple transformations.

The authors say that they significantly expand the work of Xu et al., but it doesn't appear to be the case. The choice of normalization seems to me to be the only addition that is relatively incremental. Also, even if divisive normalization is "commonly" considered in neuroscience, how this would actually be happening in this context is not clearly laid out. Is there competition or something else?
The current RNN and "training" of the model don't appear to be biologically motivated.

The paper also very quickly covers the case when the projection lays out of the manifold, which, in more complicated cases, for more extended motion, which cannot be approximated with the first order, is not solved.

**Questions:**

How does the paper concretely compare to the existing literature?
Normalization of neural activities from the action of group transformation in RNN can be thought through the learning of the Lie generators combined with retraction, which is not new.

Given that the theoretical part isn't very strong, how would the normalization be performed in a "real" neural network?

---

> ### Author Response · Authors · 2023-11-22
>
> Thank you for your thorough and valuable feedback on our manuscript. We deeply value the time and expertise you've invested in reviewing our work.

---

### Official Review · Reviewer_gVgU · 2023-11-01

**Soundness:** 4 excellent
**Presentation:** 3 good
**Contribution:** 2 fair
**Rating:** 5
**Confidence:** 2

**Summary:**

This paper studies the conditions under which grid cells form hexagonal patterns. They propose a refinement of a previously proposed theory of grid cells which states that a key ingredient in their formation is the preservation of angles (conformal property) in the mapping between physical space (R^2) and neural space (R^N). Their proposal is to normalize the vector of displacement in physical space, \delta x, by the directional derivative; they show this works in linear and nonlinear integration scenarios.

**Strengths:**

The paper builds on the excellent basis of Gao et al. (2021) to highlight conditions under which grid cells emerge. The exposition is quite clear despite the mathematical sophistication. It does offer a compelling view of grid cell emergence in light of recent skepticism about the emergence of grid cells under a broad range of training regimes (Scaeffer et al. 2022).

**Weaknesses:**

Coming from outside of this field, I was excited about recommending this paper for publication. However, I then had a look at Gao et al. (2021), and this work seems quite incremental in light of that. The scope is quite similar, most of the exposition is similar, and many of the figures look exactly the same. It's not really until the end of the paper (section 7) that the nature of the incremental advance is highlighted: 1) they removed one part of the loss (previously L_2) and absorbed it into a pre-baked normalization mechanism, and 2) they extended to the nonlinear case.

I don't know enough about this field to make a clear judgement on whether these constitute true advances, but my hunch is that these constitute minor tweaks around the edges. My biggest issue is that the justification for replacing the extra loss is that "the extra loss term is rather unnatural". If, by "unnatural", they mean "biologically implausible", I don't think replacing it with a normalization term makes this more biologically plausible; how would the brain compute directional derivates exactly? If they showed how the brain could compute this, it would definitely be a conceptual advance. It seems like in the linear scenario highlighted in 3.4, the normalization term can be calculated quite straightforwardly; they should show where each of terms come from. They should repeat the same exercise for section 3.5

My advice is that the authors should make it crystal clear in the intro how their work relates to the previous work, delineate what is old and what is new, and why this is a conceptual advance. They might consider giving a justification in terms of how the brain might be able to compute the necessary normalization; right now, there's some handwaving around Heeger-style divisive normalization, but I don't know how they would read off the key term of the directional derivative from the population in the general case.

I'm open to reversing my decision if other reviewers which are closer to this field believe this is a strong contribution.

**Questions:**

-

---

> ### Author Response · Authors · 2023-11-22
>
> Thank you for your thorough and valuable feedback on our manuscript. We deeply value the time and expertise you've invested in reviewing our work.